# Co-Infection of *Culex tarsalis* Mosquitoes with Rift Valley Fever Phlebovirus Strains Results in Efficient Viral Reassortment

**DOI:** 10.3390/v17010088

**Published:** 2025-01-11

**Authors:** Emma K. Harris, Velmurugan Balaraman, Cassidy C. Keating, Chester McDowell, J. Brian Kimble, Alina De La Mota-Peynado, Erin M. Borland, Barbara Graham, William C. Wilson, Juergen A. Richt, Rebekah C. Kading, Natasha N. Gaudreault

**Affiliations:** 1Center for Vector-Borne Infectious Diseases, Department of Microbiology, Immunology and Pathology, Colorado State University, Fort Collins, CO 80523, USA; emkate.harris@colostate.edu (E.K.H.); barb.graham@colostate.edu (B.G.); 2Center of Excellence for Emerging and Zoonotic Animal Diseases, Diagnostic Medicine/Pathobiology, Kansas State University, Manhattan, KS 66506, USAcdmcdow@vet.k-state.edu (C.M.); jricht@vet.k-state.edu (J.A.R.); 3Foreign Arthropod-Borne Animal Diseases Research Unit, United States Department of Agriculture, Agricultural Research Service, National Bio and Agro-Defense Facility, Manhattan, KS 66505, USA; james.kimble@usda.gov (J.B.K.); alina.delamota@usda.gov (A.D.L.M.-P.); william.wilson2@usda.gov (W.C.W.)

**Keywords:** Rift Valley fever phlebovirus, bunyavirus, *Culex tarsalis*, mosquito, reassortment

## Abstract

Rift Valley fever phlebovirus (RVFV) is a zoonotic mosquito-borne pathogen endemic to sub-Saharan Africa and the Arabian Peninsula which causes Rift Valley fever in ruminant livestock and humans. Co-infection with divergent viral strains can produce reassortment among the L, S, and M segments of the RVFV genome. Reassortment events can produce novel genotypes with altered virulence, transmission dynamics, and/or mosquito host range. This can have severe implications in areas where RVFV is endemic and convolutes our ability to anticipate transmission and circulation in novel geographic regions. Previously, we evaluated the frequency of RVFV reassortment in a susceptible ruminant host and observed low rates of reassortment (0–1.7%). Here, we tested the hypothesis that reassortment occurs predominantly in the mosquito using a highly permissive vector, *Culex tarsalis*. Cells derived from *Cx. tarsalis* or adult mosquitoes were co-infected with either two virulent (Kenya-128B-15 and SA01-1322) or a virulent and attenuated (Kenya-128B-15 and MP-12) strain of RVFV. Our results showed approximately 2% of virus genotypes isolated from co-infected *Cx. tarsalis*-derived cells were reassortant. Co-infected mosquitoes infected via infectious bloodmeal resulted in a higher percentage of reassortant virus (2–60%) isolated from midgut and salivary tissues at 14 days post-infection. The percentage of reassortant genotypes isolated from the midguts of mosquitoes co-infected with Kenya-128B-15 and SA01-1322 was similar to that of mosquitoes co-infected with Kenya-128B-15 and MP-12- strains (60 vs. 47%). However, only 2% of virus isolated from the salivary glands of Kenya-128B-15 and SA01-1322 co-infected mosquitoes represented reassortant genotypes. This was contrasted by 54% reassortment in the salivary glands of mosquitoes co-infected with Kenya-128B-15 and MP-12 strains. Furthermore, we observed preferential inclusion of genomic segments from the three parental strains among the reassorted viruses. Replication curves of select reassorted genotypes were significantly higher in Vero cells but not in *Culex*—derived cells. These data imply that mosquitoes play a crucial role in the reassortment of RVFV and potentially contribute to driving evolution of the virus.

## 1. Introduction

Rift Valley fever phlebovirus (RVFV) (Order: *Hareavirales*; Family: *Phenuiviridae*; Genus: *Phlebovirus*) is a zoonotic pathogen capable of causing outbreaks, resulting in substantial impact on human and livestock populations, leading to economic loss [1]. RVFV is primarily transmitted by *Aedes* and *Culex* mosquitoes or by direct contact with body fluids from infected animals [2,3,4]. More than 73 species of mosquitoes belonging to 8 genera are susceptible to RVFV, emphasizing the potential for widespread transmission through bridge vectors among naïve vertebrate populations [5,6]. Although locally acquired RVFV infection is primarily observed in sub-Saharan Africa, recent outbreaks on the Arabian Peninsula are cause for awareness of additional competent vectors and the impact RVFV transmission may have on susceptible populations [7,8]. Analysis of isolates across multiple outbreaks demonstrates high sequence similarity, suggesting favorable conditions for viral distribution to naïve populations [9,10]. Sustained mosquito-driven and transovarial transmission during epidemics can produce ideal conditions for RVFV co-circulation and co-infection of vector and vertebrate populations. The outcomes of vertebrate and vector exposure to multiple RVFV strains include emergence of novel genotypes/strains via reassortment, which could lead to increased transmission and/or virulence capable of circulation among susceptible populations. Enhanced understanding of mechanisms guiding RVFV reassortment and how complex virus/vector interactions play a role in emergence is critical to development of effective strategies to prevent and control outbreaks and the emergence of new viral strains.

Reassortment or viral segmental exchange during cellular co-infection is a hallmark amongst viruses within the order *Bunyavirales*, which possess three genome segments that are non-selectively packaged within a given host cell [11,12,13]. RVFV reassortment derived from co-infected animals, mosquitoes, or cultured cells can generate genetically distinct progeny viruses [14]. Therefore, multiple questions remain regarding RVFV and host interactions, leading to reassorted viruses. The RVFV genome consists of a Large segment (L; 6.4 kb) encoding viral RNA-dependent RNA polymerase, Medium segment (M; 3.8 kb) encoding envelope glycoproteins Gc and Gn, NSm, and the 78kDa protein, and a Small segment (S; 1.7 kb), which is ambisense and encodes a nucleoprotein (N) in the genomic-sense orientation and the non-structural protein (NSs) in the antigenomic orientation [14,15]. Infection, replication, and packaging of virions occur similarly in mammalian and insect cells, with the exception of the NSm and 78kDa proteins [14]. Mammalian models of infection indicate both proteins are non-essential in the virus life cycle; however; deficiency of NSm and 78kDa proteins reduces infection, dissemination, and transmission rate in *Culex* and *Aedes* mosquitoes [16,17]. Furthermore, virus segment incorporation into virus particles has been demonstrated to occur non-specifically, with a proportion of virions appearing to have packaged two or four segments [11,14,18,19]. Sequencing and phylogenetic analysis have documented reassortment events during epidemics where co-circulation of multiple RVFV strains/variants occurs [10]. This phenomenon combined with the relative stability of co-circulating virus strains/variants in nature produces conditions conducive for reassortment. Sheep are highly susceptible to RVFV and heavily affected during outbreaks, providing a key source for reassortment. We have previously investigated RVFV reassortment amongst strains of varying virulence using sheep to determine the frequency and genotypes of the resulting reassortant viruses. Results demonstrated virus isolated from co-infected sheep was only 1.7% of total genotypes [20]. This suggests that reassortment may occur more efficiently in mosquito populations rather than mammalian hosts.

The present study investigates RVFV reassortment in the mosquito vector using *Culex tarsalis* as a model. We determined the frequency of reassortment and characterized resulting genotypes from co-infected mosquito cells in vitro and in vivo with three different strains of RVFV. We further characterized the replication kinetics of mosquito-derived reassortants to understand implications for downstream virulence.

## 2. Materials and Methods

### 2.1. Cells and Viruses

Vero MARU (VM; Middle America Research Unit, Corozal, Panama), Vero (ATCC CCL-81, Manassas, VA, USA), and MRC-5 cells (ATCC^®^CCL-171™, Manassas, VA, USA) were cultured in complete Dulbecco’s Modified Eagle’s Medium (DMEM; Corning, New York, NY USA), supplemented with 5 or 10% fetal bovine serum (R&D Systems, Minneapolis, MN, USA; Atlas Biologicals, Fort Collins, CO, USA) and 1% antibiotic-antimycotic solution (ThermoFisher Scientific, Waltham, MA, Waltham, MA, USA) at 37 °C with 5% CO_2_ atmosphere in a cell culture incubator. *Culex tarsalis* (CxTxR2; generated at ABADRU, USDA, Manhattan, KS, USA) cells and *Aedes albopictus* C6/36 cells (ATCC^®^ CRL-1660™, Manassas, VA, USA) were maintained in Schneider’s *Drosophila* medium (ThermoFisher Scientific, Waltham, MA, USA) and Leibovitz-15 medium (ATCC-30-2008™, Manassas, VA, USA), respectively, supplemented with 10% tryptose phosphate broth (Sigma-Aldrich, St. Louis, MO, USA), 10% fetal bovine serum (IFBS, Sigma-Aldrich, St. Louis, MO, USA), and 1% antibiotic- antimycotic solution. Additionally, the CT (*Cx. tarsalis*) embryonic mosquito cell line was maintained in Schnieder’s *Drosophila* medium supplemented with sodium bicarbonate (Gibco, ThermoFisher Scientific, Waltham, MA, USA), L-Glutamine (Sigma-Aldrich, St. Louis, MO, USA), 20% heat-inactivated FBS (Atlas Biologicals, Fort Collins, CO, USA), and 1% PenStrep (ThermoFisher Scientific, Waltham, MA, USA) [21]. 

Rift Valley fever phlebovirus strains Kenya-128B-15 (Ken06; GenBank: KX096938, KX096939, and KX096940), SA01-1322 (SA01; Genbank KX096941, KX096942, and KX096943), and ZH501 (GenBank: DQ380149, DQ380200, and DQ375406) strains were amplified in C6/36 cells or Vero cells [22,23,24]. The live attenuated strain MP-12 strain (Genbank: DQ375404, DQ380208, and DQ380154) was propagated in MRC-5 for in vitro or Vero cells for in vivo infection [25,26]. All virus-containing materials (cell culture supernatants, and tissue homogenates) were titrated by a standard plaque assay as described previously and below for each assay [27].

### 2.2. In Vitro Co-Infection of RVFV Strains to Assess Reassortment Profiles Using CxTxR2 Cells

Assessment of in vitro reassortment was completed using CxTxR2 cells co-infected at a MOI of 0.1 with RVFV Kenya-128B-15 and MP-12 or Kenya-128B-15 and SA01-1322 strains, respectively. Cells were incubated for 1 h at 28 °C. Unadsorbed virus was removed by rinsing cells twice with growth medium, followed by addition of 0.2 mL of Drosophila S2 growth medium. Cells were incubated at the above-described conditions for 3 days, whereupon supernatant was collected and stored at −80 °C for further analysis.

### 2.3. Mosquitoes, Infection with RVFV Strains, Co-Infection, and Blood Meal Titration

For all experiments, *Cx. tarsalis* (strain KNWR) mosquitoes were reared and held in a controlled environment at 28 °C, 70% humidity, and a 16:8 light:dark cycle. Approximately 12–16 h prior to receipt of infectious bloodmeal, 50 female mosquitoes (7–10 days old) were aspirated into cartons and starved of sugar and water. Infectious bloodmeal was prepared by harvesting RVFV-infected Vero cell supernatant at 72 h post-infection (hpi) and clarified by centrifugation at 7000× *g* for 10 min. Virus was then mixed in a 1:1:1 ratio of either Kenya-128B-15 and MP-12 or Kenya-128B-15 and SA01-1322 with fresh defibrinated calf blood (Colorado Serum Company, Denver, CO, USA), loaded onto a Hemotek membrane feeder, and (Hemotek Ltd., Hampshire, UK) covered with parafilm [28]. Membrane feeders attached to a heating unit were warmed to 37 °C and applied to cartons containing starved mosquitoes for approximately 1 h after which blood-fed females were transferred to new cartons. At 14 days post-infection (dpi), mosquito legs, wings, midguts, and salivary glands were removed (*n* = 30) and pooled into tissue-specific tubes containing mosquito diluent (DMEM containing 20% FBS, 1% penicillin, and streptomycin, 0.1% gentamicin, and 0.1% fungizone) and two glass beads, as previously described [11,12]. Tissues were homogenized via Tissue Lyser LT (Qiagen, Germantown, MD, USA) and stored at −80 °C prior to plaque isolation. Viral bloodmeal titer was determined via neutral red plaque assay as previously described [29,30] (Appendix A). Briefly, 10-fold dilutions of individual clarified virus or co-infected blood meal were used to infect confluent Vero cells seeded to 6-well plates (CellTreat, Pepperell, MA, USA) and rocked for 1 h, upon which a solution containing 0.4% agarose (Lonza Rockland, Rockland, ME, USA) dissolved in supplemented growth media was added to cells. On day three, a solution of 0.33% neutral red (Sigma Aldrich, St. Louis, MO, USA) and 0.4% agarose in supplemented growth media was added to the first overlay. The second overlay was incubated for a minimum of 3–4 h prior to enumeration and calculation of plaque-forming units (PFU) per mL.

### 2.4. Reassortant Virus Isolation by Plaque Purification

Parental and reassortant viruses (RAVs) were isolated from co-infected CxTxR2 cells and *Culex*-infected mosquito tissues by plaque purification. Briefly, dilutions of virus samples (mosquito tissue homogenates or cell culture supernatants) were added to confluent monolayers of Vero MARU cells in 6-well cell culture plates (Corning, New York, NY, USA). Samples were incubated on cells for an hour at 37 °C with 5% CO_2_, removed and replaced with a 0.9% agarose-medium overlay containing equal volumes of 1.8% agarose in distilled water (Fisher Scientific, Waltham, MA, USA) and 2x minimum essential medium (MEM; Fisher Scientific, Waltham, MA, USA), supplemented with 10% fetal bovine serum and 2% antibiotic-antimycotic solution. Plates were incubated at 37 °C with 5% CO_2_ for 2–3 days, whereupon plaques were stained with a mixture of neutral red (Sigma Aldrich, St. Louis, MO, USA) agarose overlay medium and incubated for a minimum of 3–4 h. Plaques were isolated and transferred onto Vero MARU cells seeded in 48 well plates, where virus was amplified for three days, followed by supernatant collection, and storage at −80 °C for further analysis.

### 2.5. Genotyping of Plaque Purified Viruses

Viral RNA was extracted from cell culture supernatants using QIAmp Viral RNA Mini kit (Qiagen, Germantown, MD, USA), according to the manufacturer’s instructions, and stored at −80 °C. A previously described One-step RT-qPCR genotyping assay was performed to determine RVFV segment composition as previously described [31]. Briefly, the genotyping assay primers were mixed with 10 µL of q-script XLT 2× mix (Quantabio, Beverly, MA, USA), 1 µL of Eva green (20×) (Biotium, Fremont, CA, USA), 2.5 µL of RNA template and nuclease-free water up to a total volume of 20 µL. The amplified PCR products were subjected to one round of melt curve analysis with increasing temperatures from 70 to 95 °C at the rate of 0.20 °C change every 10 s. Melt curves were analyzed using CFX 3.0 software (Bio-Rad, Hercules, CA, USA). Based on the peak melting temperature, the samples were categorized as Kenya-128B-15 or MP-12 or SA01-1322 strains L or M or S segment. Moreover, the strain identities of genotyped plaques (approx. 10%) were confirmed by Sanger sequencing [31].

### 2.6. Replication Kinetics of Parental and Reassorted Genotypes

Multi-step growth curves were generated for RVFV strains Kenya-128B-15, ZH501, and SA01-1322 by infecting confluent 25cm^2^ flasks (Celltreat, Pepperell, MA, USA) of Vero or CT cells (Appendix A). Flasks were infected in triplicate at an MOI 0.01 for 1 h after which media was decanted, cells rinsed with 1× PBS (Gibco), and provided fresh supplemented media. Supernatant representing infection time points was removed at 0, 3, 6, 12, 24, 36, 48, 60, and 72 for Vero and, additionally, at 108, and 120 hpi for CT cells. Viral titer was quantified by calculating the PFU/mL for each sample by neutral red plaque assay on Vero cells as described above.

Replication kinetics of RVFV Kenya-128B-15, MP-12, and SA01-1322 and selected mosquito-derived reassortant RVFV isolates was performed using CxTxR2 and VM cells seeded onto 48-well plates and infected at MOI 0.01. At 1 hpi, virus-inoculated medium was replaced with 200 µL of respective complete medium. Virus supernatant was collected at 24, 48, 72, 96, and 120 hpi (CxTxR2 cells) or 12, 24, 36, 48, and 72 hpi (VM cells) and titered by plaque assay. Virus titer was determined via plaque assay using VM cells seeded onto 24-well plates to a density of 1 × 10^5^ cells per well and incubated overnight at 37 °C with 5% CO_2_. On the day of infection, serial dilution of virus sample was prepared in complete DMEM and added onto VM cells [27]. At 1 hpi, the infection medium was replaced with overlay containing 1% methylcellulose-1x MEM (ThermoFischer Scientific, USA), 5% FBS, and 1% antibiotics/antimycotic solution. After five days post-infection, cells were fixed with 5% crystal violet fixative solution, and PFU/mL calculated.

### 2.7. Statistical Analysis

Raw virus growth curve data were log-transformed (base 10) to stabilize variance prior to analysis. Comparisons were only made between RAVs and the parental strains. Growth curves were compared using a pooled two-sample t-test between groups (all pairwise comparisons) for all time points. Mean t-values were evaluated using permutation tests (10,000 simulations), where samples were randomly placed into each of the two groups and mean t recalculated [32]. *p*-value is expressed as a proportion of permutations where the absolute value of the statistic is greater than the mean t for the original data. *p*-values were adjusted for multiple comparisons using the Holm method. Analysis was completed using R version 4.3.0 (2023-04-21 ucrt) using the *statmod* package version 1.5.0 [33].

### 2.8. Regulatory Compliance

All work involving virulent RVFV strains Kenya-128B-15 and SA01-1322 was performed at high containment BSL3+ facilities at the Biosecurity Research Institute (BRI) at Kansas State University (KSU) or Colorado State University (CSU) in compliance with USA regulations and conducted with KSU IBC approval #1544.11 and CSU IBC approval #20-024B.

## 3. Results

### 3.1. In Vitro RVFV Co-Infection of Cx. tarsalis Cells Demonstrates Low Levels of Reassortment

To characterize reassortment frequency of segmental exchange between a virulent strain, Kenya-128B-15, and a live attenuated vaccine strain, MP-12, co-infection of CxTxR2 cells was conducted (Figure 1A). Genotyping of plaques isolated from the infected cell supernatant indicated the majority of recovered viruses were parental Kenya-128B-15 (58.5% or 24/41 plaques) and MP-12 (39.1% or 16/41 plaques) strains (Figure 1B). The singular reassortant genotype recovered from co-infected CxTxR2 cells consisted of the L segment from parental Kenya-128B-15 with S and M segments from MP-12 and represented only 2.4% (1/41) of isolated plaques. Similarly, another co-infection was performed with two virulent RVFV strains: Kenya-128B-15 and the Saudi Arabian strain, SA01-1322. Of the 44 plaques isolated, 97.7% (43/44) consisted of parental Kenya-128B-15 and none represented parental SA01-1322. Reassortment between the L and S segments from Kenya-128B-15 and M segment of SA01-1322 was detected in only one of the plaques analyzed (2.3% or 1/44). These data demonstrate that in vitro co-infection of diverse RVFV strains in *Cx. tarsalis* cells result in low frequency of RAV.

### 3.2. In Vivo RVFV Co-Infections in Cx. tarsalis Mosquitoes Result in Frequent Reassortment That Is Detected in Midgut and Salivary Gland Tissue

*Culex tarsalis* mosquitoes (n = 30) were co-infected with the same RVFV strain combinations as used in in vitro experiments described above and RAVs were isolated from midgut and salivary tissue collected at 14 dpi (Figure 2A). Viral genotypes isolated from Kenya-128B-15 and MP-12 co-infected mosquitoes resulted in 40% (36/90) recovery of parental Kenya-128B-15 in the midgut and none from salivary gland tissues (Figure 2B). Additionally, 13% (12/90) and 46% (41/90) of plaque isolates recovered from the midgut and salivary gland tissues, respectively, represented parental MP-12. Reassortant genotypes comprised approximately 47% (42/90) of isolated plaques in the midgut: (1) Kenya-128B-15_LS_:MP-12_M_ (19%; 17/90); (2) Kenya-128B-15_S_:MP-12_LM_ (16%; 14/90); (3) Kenya-128B-15_L_:MP-12_MS_ (11%; 10/90); and (4) Kenya-128B-15_MS_:MP-12_L_ (1%; 1/90) (Figure 2B). Only Kenya-128B-15_LS_:MP-12_M_ (13%; 12/90) and Kenya-128B-15_S_:MP-12_LM_ (41%; 37/90), which represented the highest percentage detected in the midgut, were further isolated from salivary tissue (Figure 2B).

Further co-infection experiments using two virulent strains of RVFV (Kenya-128B-15 and SA01-1322) resulted in 31.5% (29/92) and 92% (47/51) of plaques isolated from the midgut and salivary glands of *Cx. tarsalis*, respectively, representing parental Kenya-128B-15 (Figure 2C). Furthermore, parental SA01-1322 represented 8.7% (8/92) plaques from midgut and 6% (3/51) salivary glands. RAVs detected in midgut tissue comprised approximately 60% (55/90) plaques isolated and consisted of the following: (1) Kenya-128B-15_LS_:SA01-1322_M_ (23.9%; 22/92); (2) Kenya-128B-15_S_:SA01-1322L_M_ (23.9%; 22/92); (3) Kenya-128B-15_MS_:SA01-1322_L_ (9.8%; 9/92); and (4) Kenya-128B-15_M_:SA01-1322_LS_ (2.2%; 2/92) (Figure 2C). Only one of the 51 plaques isolated from salivary gland represented a reassortant genotype: Kenya-128B-15_M_:SA01-1322_LS_ (2%) (Figure 2C).

Follow-up virus isolation was completed using pooled legs and wings (n = 30) of mosquitoes infected with Kenya-128B-15 and SA01-1322 to assess potential midgut escape barriers for RAV detected in the midgut but not salivary glands of co-infected mosquitoes. Of the 48 plaques isolated from legs and wings, 100% represented the parental SA01-1322 strain (data not shown).

A higher percentage of reassortant genotypes were detected in the salivary glands during co-infection with Kenya-128B-15 and MP-12 than mosquitoes co-infected with Kenya-128B-15 and SA01-1322. Overall, the majority of RAV isolated across all combinations of RVFV co-infections contained the L and M segments from one parental strain and the S of the other (Figure 3A,B).

### 3.3. Replication Kinetics of RVFV Reassortants Do Not Display Overt Replicative Advantage Compared to Parental Strains

To determine replicative titers of the reassortant genotypes compared to parental strains, in vitro growth curves were performed in Vero MARU (VM) and CxTxR2 cells. Reassortant virus isolates that represented the highest frequency genotypes in midgut tissue were amplified in CxTxR2 cells for further analysis and consisted of: (1) Kenya-128B-15_LS_:MP-12_M_; (2) Kenya-128B-15_S_:MP-12_LM_; (3) Kenya-128B-15_LS_:SA01-1322_M_; and (4) Kenya-128B-15_S_:SA01-1322_LM_. Infection with RAV Kenya-128B-15_LS_:MP-12_M_ in VM and CxTxR2 cells demonstrated significantly slower replication kinetics compared to parental Kenya-128B-15 (*p* = 0.03 and *p* = 0.001, respectively; Figure 4A,B). Although not significant, RAV Kenya-128B-15_LS_:MP-12_M_ also had reduced replication compared to parental MP-12 in VM cells (*p* = 0.18) but showed significantly lower titers in CxTxR2 cells (*p* = 0.001) (Figure 4A,B). Reassortant Kenya-128B-15_S_:MP-12_LM_ exhibited similar titers as that of both parental strains and reached similar peak titers of 10^6^ PFU/mL when grown in VM cells (Kenya-128B-15 *p* = 0.95; MP-12 *p* = 0.54) (Figure 4B). Replication kinetics of RAV Kenya-128B-15_S_:MP-12_LM_ in CxTxR2 cells was significantly lower than both parental strains (Kenya-128-B15 *p* = 0.003; MP-12 *p* = 0.03); however, viral titers were similar by 120 hpi (Figure 4B).

Replication efficiency of RAV Kenya-128B-15_LS_:SA01-1322_M_ was significantly higher compared to parental Kenya-128B-15 (*p* = 0.01) and SA01-1322 (*p* = 0.0009) strains in VM cells (Figure 5A). Likewise, RAV Kenya-128B-15_S_:SA01-1322_LM_ also had significantly higher replicative ability in VM cells relative to Kenya-128B-15 (*p* = 0.0009) and SA01-1322 parental strains (*p* = 0.0006) (Figure 5A). In contrast, in CxTxR2 cells, replication curves of both RAVs were statistically similar to parental Kenya-128B-15 (RAV Kenya-128B-15_LS_:SA01-1322_M_ *p* = 0.13; RAV Kenya-128B-15_S_:SA01-1322_LM_ *p* = 0.15), demonstrating low replication efficiency in this cell line. However, when compared to SA01-1322 strain in CxTxR2, RAV Kenya-128B-15_LS_:SA01-1322_M_ (*p* = 0.009) and RAV Kenya-128B-15_S_:SA01-1322_LM_ (*p* = 0.03) demonstrated a significantly lower replication efficiency prior to reaching similar titers at 120 hpi (Figure 5B).

These data show that parental Kenya-128B-15 displays a replicative advantage over RAVs isolated from Kenya-128B-15 and MP-12 co-infected mosquitoes in mammalian and mosquito cells that is not compensated for by segmental contribution from MP-12. However, certain RAVs isolated from mosquitoes co-infected with Kenya-128B-15 and MP-12 strains displayed replicative advantage in mammalian cells as compared to mosquito cells. Furthermore, RAVs from Kenya-128B-15 co-infection with SA01-1322 showed increased replication in mammalian cells, which was not recapitulated in mosquito cells.

## 4. Discussion

Rift Valley fever phlebovirus is an emerging mosquito-borne virus that poses a threat to human and veterinary health. The tri-segmented genomic composition is capable of segmental reassortment between co-circulating RVFV strains, producing novel reassortant genotypes with increased transmission and/or pathogenicity. However, the frequency of reassortment and its impact on the ecology of the virus are unclear, particularly in the context of the mosquito vector. Field- and laboratory-generated data suggest RVFV reassortment arises in co-infected cells of vertebrate hosts and/or mosquitoes [14,34]. Because mosquitoes possess the capacity to acquire multiple infectious bloodmeals, their ability to drive RVFV outbreaks and become infected with multiple viral strains is significant. Here, we investigated mosquito-driven dynamics of RVFV reassortment in a *Culex tarsalis*-derived cell line and blood-fed mosquitoes. Three parental strains of RVFV were utilized to investigate reassortment outcomes between a virulent and vaccine strain (Kenya-128B-15 and MP-12) or two virulent strains (Kenya-128B-15 and SA01-1322). Our data demonstrate that, although reassortment was infrequent (<2%) in vitro, the number of reassortant viruses recovered upon co-infection in vivo increased to approximately 2-60% across mosquito midgut and salivary glands. Further analysis of growth kinetics of two of the most frequently isolated RAVs derived from Kenya-128B-15 and MP-12 co-infected mosquitoes demonstrated a replicative disadvantage compared to parental strains in mammalian and mosquito cells, despite isolation directly from mosquito midgut and salivary glands. Reassortant viruses isolated from co-infection with Kenya-128B-15 and SA01-1322, which replicated to higher titers in mammalian cells compared to parental strains, were only isolated from the mosquito midgut but not from the salivary glands. Therefore, although reassortment between two parental strains readily occurred in mosquitoes, phenotypic characterization did not demonstrate enhanced replication over parental virus using in vitro approaches. Altogether, these data illustrate the complexity of reassortment among bunyaviruses, particularly within highly similar strains of RVFV, and key biological factors unique to the mosquito vector.

Previous work has demonstrated that in vitro co-infection between bunyaviruses varies in terms of reassortment prevalence [35]. In this study, we used strains of RVFV with significant genomic similarity to investigate segmental exchange from co-infected CxTxR2 cells. At 3 dpi, approximately 2% of plaques isolated from RVFV co-infected cell cultures represented reassortant genotypes, regardless of strains used. The majority of viruses (i.e., 57–97%) isolated post-infection consisted of parental genotypes. These outcomes could be due to low levels of individual cells co-infected with different virus strains. Additionally, it is possible that environmental conditions present in vitro are not efficient to produce reassortment. Ly et al. conducted a study where C6/36 cells were infected with RVFV MP-12 and Arumwot virus (AMV) or Gouleako virus (GOLV), which resulted in a complete lack of reassortment [35]. Their results demonstrated that reassortment between RVFV and other bunyaviruses does not readily occur. However, in that same study, nearly 83% reassortment was observed in C6/36 cells co-infected with MP-12 and rMP12-GM50, a recombinant virus containing silent mutations across open reading frames for the N, NSs, M, and L genes [35]. Viral replication efficiency of one parental strain over another may impact reassortant outcomes. Heitmann et al. demonstrated replicative advantage of *Bunyamwera orthobunyavirus* (BUNV) over *Batai orthobunyavirus* (BATV) when both viruses were inoculated at a 1:1 ratio, independent of host cell background [36]. An increase in reassortment was observed in BHK-21 cells when infectious inoculum for BATV was 50 times higher than BUNV (MOI of 5 versus 0.1) [36]. Future work can focus on altering viral infection dynamics in vitro to observe downstream patterns of reassortment. Our previous and Appendix A do not indicate a significant difference in the replication kinetics among the three strains used in our studies, supporting the use of equivalent MOIs for co-infection of the cell cultures (Appendix A) [30]. We selected a lower MOI to allow for multiple rounds of viral replication to increase the likelihood of co-infection by 3 days post-infection without significant cell death [20]. Our in vitro data suggest that RVFV reassortment is not efficient in CxTxR2 cells, even though no significant difference in replication was observed between the strains used in this study; however, using different MOIs may alter the pattern and prevalence of RAV genotypes. Ultimately, in vitro co-infection experiments, although a convenient system, provided a narrow perspective regarding RVFV reassortment. Alternatively, our in vivo approach produced more assessable and valuable data in regard to RVFV infection replication, dissemination, and reassortment.

Because previous data suggest that *Culex* mosquitoes provide a good model for RVFV infection and viral reassortment, we chose *Cx. tarsalis* mosquitoes to expand our in vitro data [28,34,37,38]. At 14 dpi, with a co-infected RVFV bloodmeal, we observed a high percentage of RAV genotypes in *Cx. tarsalis* midgut and salivary glands. Mosquitoes exposed to RVFV Kenya-128B-15 and MP-12 strains resulted in 47% and 54% RAV genotypes recovered from the midguts and salivary glands, respectively. Among the reassortant viruses, two genotypes were predominantly detected in both midgut and salivary gland tissues: Kenya-128B-15_LS_:MP-12_M_ and Kenya-128B-15_S_:MP-12_LM_. Although these RAVs could have arisen independently in both tissues, these data most likely indicate viral dissemination from the midgut to the salivary glands for potential transmission. Expectorated saliva was not assessed in this study but should be the focus of future work to determine how efficiently RAVs can overcome the salivary gland escape barriers compared to parental strains. Replication of the RAV Kenya-128B-15_LS_:MP-12_M_ was significantly lower than parental strains, while RAV Kenya-128B-15_S_:MP-12_LM_ showed no significant decrease in replication in mammalian- or *Cx. tarsalis*-derived cells. This could indicate decreased fitness phenotypes and transmission profiles in mosquitoes for these RAVs compared to parental viruses.

Further assessment of RVFV reassortment between two virulent strains in *Cx. tarsalis* using Kenya-128B-15 and SA01-1322 produced reassortants in 60% of virus plaques isolated from midguts. Nearly all RAV from Kenya-128B-15 and SA01-1322 co-infected mosquitoes were restricted to the midgut (55/92 plaques in midgut versus 1/51 in salivary tissue), a striking difference from co-infection with Kenya-128B-15 and MP-12 strains where reassortment was observed at near similar frequency in midgut and salivary tissues. Viral titration of input virus revealed a higher load of Kenya-128B-15 compared to SA01-1322. This may account for representation of Kenyan parental strain and overall segments recovered in mosquito tissues. Additionally, viral dissemination determined by assessment of infected mosquito legs and wings resulted in the detection of only parental SA01-1322. This demonstrates that both parental strains, although inoculated at different titers, were able to establish infection and downstream dissemination. Previous research by Beaty et al. demonstrated that midgut escape of reassortant virus derived from La Crosse (Order: *Elliovirales*; Family: *Peribunyaviridae*: Genus: *Orthobunyavirus*) and snowshoe hare (Order: *Elliovirales*; Family: *Peribunyaviridae*; Genus: *Orthobunyavirus*) virus were diminished for those genotypes containing either LAC or SSH S segments [39,40]. The RVFV strains used here have high amino acid similarity across all segments; thus, reassorted genotypes with different segmental combinations may have less impact on tissue escape and dissemination. However, it is possible that selective pressure on reassortant strains in the midgut limited dissemination compared to parental strains. Additional work should investigate viral interactions with mosquito tissue escape barriers or alternative mechanisms for dissemination which may govern this result [41]. Experiments wherein viral MOI can be altered (as mentioned above) and those that include serial exposure to infectious bloodmeals should be undertaken. Such experiments will help to determine if superinfection exclusion plays a role in reassortment frequency and whether this occurs focally or across the entire tissue. Further, reassortment frequency and restriction derived from co-infection in alternative mosquito species, such as *Aedes aegypti*, should also be explored. The majority of reassortant genotypes recovered represented Kenya-128B-15_LS_:SA01-1322_M_ and Kenya-128B-15_S_:SA01-1322_LM_. Overall viral titer and replication of both RAVs in Vero-MARU were nearly half a log higher than parental strains but statistically similar to parental viruses in *Culex*-derived cells. These data may indicate decreased fitness and lack of ability to overcome escape barriers in the mosquito. Not investigated in the current study is the interferon response, which is suppressed by NSs during the RVFV replication. Vero cells are deficient in interferon expression, which could have impacted the replication kinetics of the RAV investigated in this study [42]. It is of interest and the aim of future studies to assess the replication kinetics of the RAVs in interferon competent RVFV susceptible cell lines such as Baby Hamster Kidney (BHK-21) and *Aedes albopictus* (C6/36 and/or U4.4) cells [36]. In our previous work, we used Madin-Darby ovine kidney (MDOK) cells to assess reassortment and this tool would fit well within the repertoire of resources used in the present work [20]. The primary genotype (>92%) of viral genotypes recovered from salivary glands comprised parental Kenya-128B-15. The only reassortant genotype detected in salivary tissue was the product of the M-segment from Kenya-128B-15 packaged with the L- and S-segments from SA01-1322. Previous literature suggests the M segment (NSm specifically) is critical to RVFV infection and dissemination from the mosquito midgut [17]. Differential susceptibility of mosquitoes to wild-type RVFV strains and a potential genetic basis for this observation involving the M segment warrant further investigation beyond the scope of this study.

Overall, our in vivo data are congruent with previous work examining reassortment between RVFV strains ArD38661 and ZH501 in *Culex pipiens* mosquitoes [34]. Turell et al. assessed individual whole mosquitoes post-blood feeding and observed that nearly all 41 viral plaques produced from three infected mosquitoes consisted of RAVs [34]. A limitation of the data generated by Turell et al. includes using a co-infected hamster as the source of the infectious blood meal; thus, it is possible that reassortment occurred in the mammalian host prior to ingestion by the mosquito [34]. Furthermore, only two segments (M and S) were assessed using antibodies raised against strain-specific segments. A limitation to data generated in the current study is the possibility that certain combinations of reassortants not detected with in vitro and in vivo experiments could display a slower rate of replication and/or fitness. Thus, plaque isolation of such phenotypes would not have occurred with the methods utilized here. This could be defined in future studies by assessing plaque isolation from co-infected tissue beyond the standard protocol timeline. Despite this, the data generated from our studies provide comprehensive information on RVFV reassortment regarding all segments for each of the three strains used with a focus on tissue-specific localization that will serve as a foundation for future investigations.

Reassortment among bunyaviruses is well documented and can serve to inform the current work. Analysis of 99 bunyaviruses indicated that reassortment occurred with relative frequency, oftentimes including the L- and S- segments of one viral strain and the M-segment of the other co-infecting virus [43]. This pattern seems to predominate because the N protein encoded on the S segment, along with the RNA polymerase from the L segment, are a critical part of viral replication and, therefore, co-evolved together [44]. However, current data also suggest that RVFV packaging is non-selective [11,14]. Emergence of Iquitos, Itaya, Schmallenberg, and Ngari virus via reassortment all depict a predicted pattern of L- and S- segments contributed from one virus and the M- segment from a second bunyavirus. Our data demonstrate an alternative trend where the majority of RAV genotypes isolated consisted of the L- and M- segment from one parental strain and the S of the other. This may be more similar to observations made for La Crosse virus reassortment, where segmental exchange was more stochastic [45]. Future experiments will use the tools established here to further interrogate robustness of the observed patterns and potential contributions by proteins expressed on specific segments that may have downstream implications for RAVs detected in the field. For example, Bird et al. showed that nonsynonymous mutations across each RVFV segment were low, highlighting the importance of genome stability used by the virus to maintain infectivity in mammalian and mosquito hosts [24].

Altogether, the results provided here allow insights into RVFV reassortment within a relevant mosquito model. While bunyaviral reassortment is well documented, our understanding of RVFV segmental exchange among closely related strains requires more research. We report a system for viral infection in mosquitoes and mosquito cells that can be used to assess multiple aspects of RVFV reassortment and downstream implications on replication kinetics and virus dissemination. Future studies will include characterization of transmission patterns of RAVs in mosquitoes. The work reported here fills an important gap to enhance our knowledge regarding RVFV reassortment and dissemination in a competent mosquito vector, which contributes to our understanding of the emergence of novel RVFV strains and will aid in developing effective countermeasures.

## Figures and Tables

**Figure 1 viruses-17-00088-f001:**
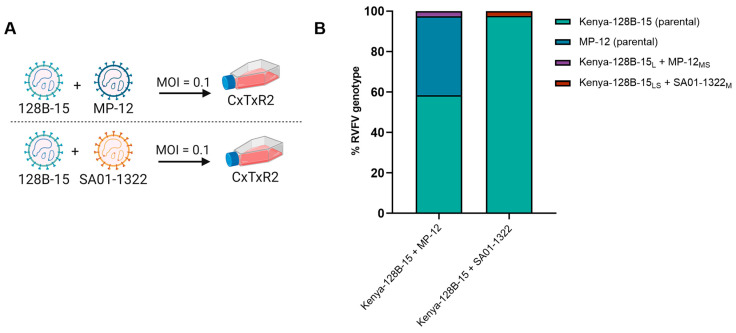
In vitro co-infection of *Cx*. *tarsalis* cells (CxTxR2) with two strains of RVFV results primarily in recovery of parental Kenya-128B-15 virus strain with low frequency of RAVs detected. CxTxR2 cells were co-infected at 0.1 MOI with RVFV Kenya-128B-15 and MP-12 (**A**) or Kenya-128B-15 and SA01-1322 (**B**). Virus supernatant was collected at 3 days post-infection (dpi) and virus was plaque purified for genotyping analysis to determine segmental composition. Results are based on one independent experiment.

**Figure 2 viruses-17-00088-f002:**
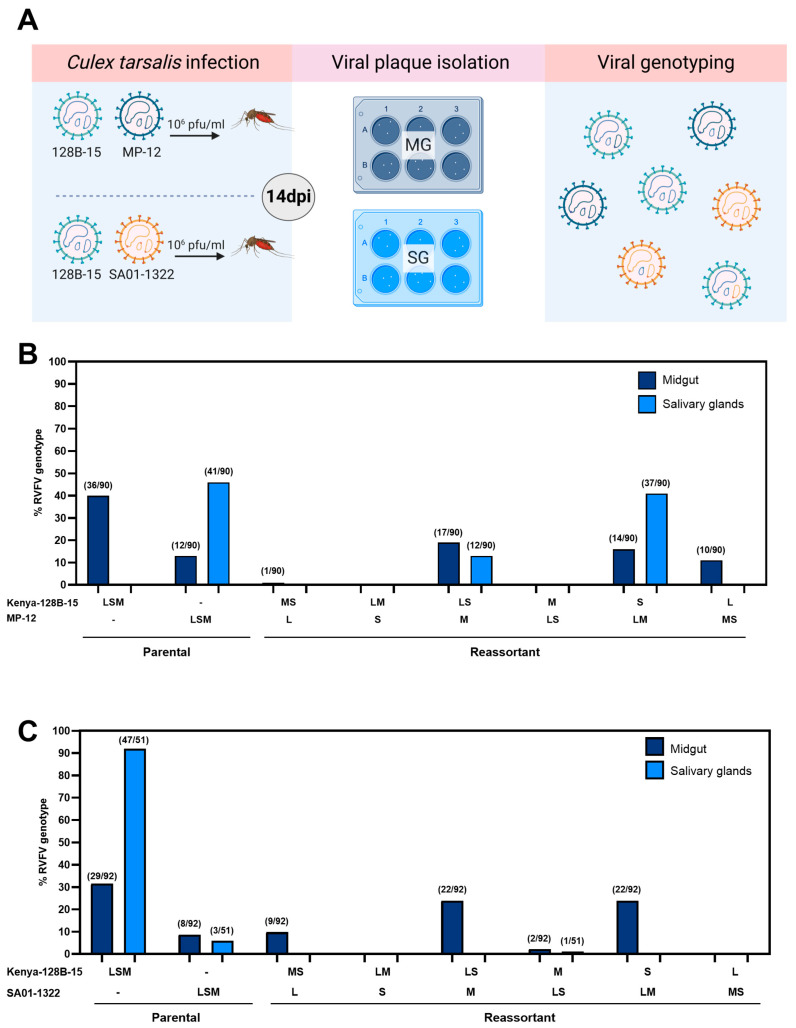
Rift Valley fever virus co-infection in *Cx. tarsalis* produces viral genotypes representing parental and reassortant strains across mosquito midgut and salivary gland tissue. Adult female mosquitoes were provided an RVFV infectious bloodmeal containing either Kenya-128B-15 and MP-12 or Kenya-128B-15 and SA01-1322. At 14 dpi, midgut (dark blue) and salivary glands (light blue) were dissected from female mosquitoes (n = 30) and pooled into tissue-specific tubes. The virus was isolated from tissue-specific homogenates and genotyped to determine segmental composition (**A**). Percent genotyped virus on the *y*-axis with segmental composition on the *x*-axis. The number of each genotype detected over the total number of plaques analyzed from the midgut and salivary gland tissues of Kenya-128B-15 and MP-12 (**B**) Kenya-128B-15 and SA01-1322 (**C**) co-infected mosquitoes are shown. Data are representative of two independent experiments.

**Figure 3 viruses-17-00088-f003:**
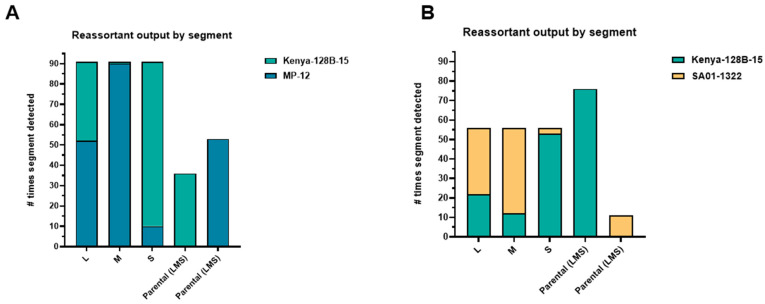
Analysis of segmental composition in isolated RAV genotypes from infected *Cx. tarsalis* reveals patterns of reassortment present in the majority or minority of isolated viruses. Raw counts of the L-, M-, or S -segment recovered from *Cx. tarsalis* midgut and salivary gland tissue were totaled and grouped by co-infection with either Kenya-128B-15 and MP-12 (**A**) or Kenya-128B-15 and SA01-1322 (**B**).

**Figure 4 viruses-17-00088-f004:**
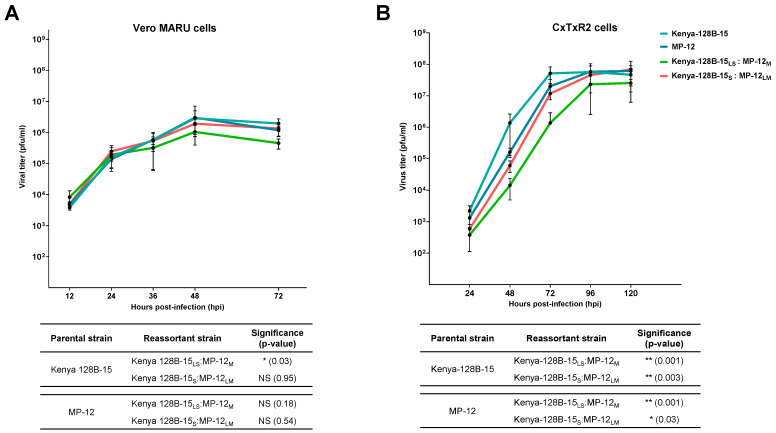
Growth curves of frequently detected reassortant RVFV genotypes isolated from co-infected *Cx. tarsalis* compared to parental strains reveals comparable or decreased overall growth curves across cell types. Growth kinetics of parental and RAVs (Kenya-128B-15_LS_:MP-12_M_ and Kenya-128B-15_S_:MP-12_LM_) in Vero MARU (**A**) and CxTxR2 (**B**) cells. An MOI of 0.01 was used for infection and supernatant collected post-infection. The mean growth curve for each reassortant strain was compared to parental using a two-sample *t*-test. Data with *p* < 0.05 were considered significant. All data are indicative of two independent experiments.

**Figure 5 viruses-17-00088-f005:**
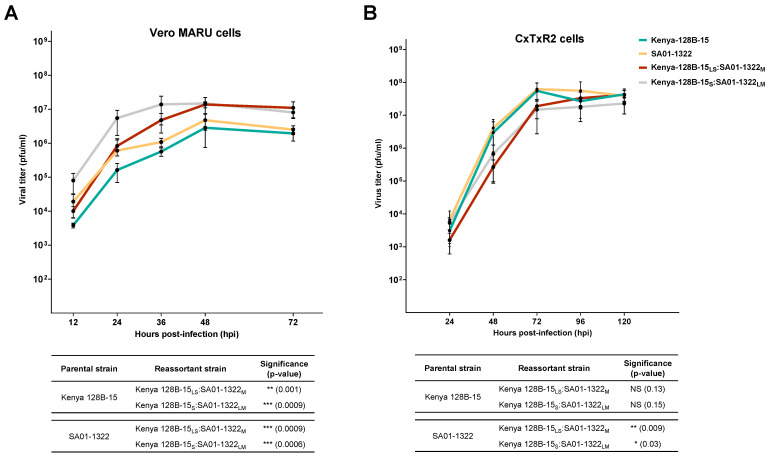
Reassortant viruses generated by co-infection between Kenya-128B-15 and SA01-1322 replicated to higher titers than parental strains in Vero MARU cells but not CxTxR2 cells. Growth curves of the parental strains and the two most frequently recovered RAVs isolated from co-infected *Cx. tarsalis* were analyzed in Vero MARU (**A**) and CxTxR2 (**B**) cells. Each viral strain was inoculated into respective cell lines at an MOI of 0.01 and supernatant collected post-infection. The mean growth curve for each reassortant strain was compared to parental using a two-sample *t*-test. Data with *p* < 0.05 were considered significant. All data are indicative of two independent experiments.

## Data Availability

Data are contained within the article and Appendix A.

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
