# Peer review of "Co-Infection of *Culex tarsalis* Mosquitoes with Rift Valley Fever Phlebovirus Strains Results in Efficient Viral Reassortment"

_viruses, 2025, doi:10.3390/v17010088_

Round 1
Reviewer 1 Report
Comments and Suggestions for Authors
Comments on Harris et al., “Co-infection of Culex tarsalis mosquitoes with RVFV strains results in efficient viral reassortment”.
This is an impactful manuscript describing relative rates of RVFV genomic segment reassortment in mosquito cells in vitro and in tissue types in whole animals. The authors have followed up on previous in vivo experimentation to examine reassortment of viral segments from different strains of RVFV when sheep are co-infected with multiple strains. Here, they use mosquito cells and live mosquitoes to show that rates of reassortment are dramatically higher in specific tissue types in live mosquitoes than in insect cells in culture or in mammalian cells. While there are several caveats and limitations to this study, it provides a new appreciation for propensity for genome segment reassortment in different host/vector species and in different tissue types within the mosquito.
It should be noted that in a recent development in the viral taxonomic structure, the old “Bunyavirales” Order has been elevated to a Class called the “Bunyaviricetes”, with two new Orders. The relevant Order for RVFV is now “Hareavirales” and the family is the “Phenuivirdae” (https://doi.org/10.1128/jvi.01069-24). The authors should make the appropriate corrections on Lines 43 and 44.
There is a mistake in describing the coding strategy of the ambisense S segment of RVFV. It says, starting on line 69: “…Small segment (S; 1.7kb) encoding a nucleoprotein (N) in the sense orientation and the non-structural protein (NSs) in the antisense orientation.” In fact (as the authors surely know), the genomic RNA on the S segment is in the negative sense (antisense) with respect to N, i.e., when transcribed you get the + sense mRNA. The NSs gene is in the positive sense in the genomic RNA, which needs to be replicated into the antigenome before transcription of the NSs gene into the + sense mRNA. This should be clarified.
Although the experimental design is solid and the results largely support the conclusions, one aspect that warrants more discussion and future experimentation is the lack of clear discrimination between the likely drivers of reassortment versus ‘non-reassortment’. That is, reassortment absolutely requires co-infection of individual cells. On the other hand, non-reassortment could be the result of either absence of co-infection of individual cells, or some other host cell restriction of certain combinations of segments or simply that certain reassortants exhibit significantly reduced fitness in at least one cell type in the host.
Therefore, it is somewhat puzzling why a low MOI was chosen for the experiments in cell culture. It is strictly possible that so few cells were productively infected (with both viruses) in the cell culture experiments that reassortment did not have the opportunity to happen, while infection dynamics in a live mosquito midgut would have been more conducive numerically to coinfection and thus reassortment. On the other hand, if there were host cell type specific restriction mechanisms for which certain components of the virus strain specific segments provided a means for viral escape, this could be deduced in further in vitro and in vivo viral replication experiments.
Along these same lines, although it is somewhat obvious that the authors used Vero cells for the mammalian cell infections to maximize replication/propagation, since these cells do not mount an interferon defense to infection and the RVFV countermeasures to interferon pathway activation is in S or distributed between S and M, it seems a missed opportunity that only these mammalian cells were used. The authors should discuss this, and probably put it on the list of experiments to do for next time!
Overall this is a good and important work.
Author Response
Comments on Harris et al., “Co-infection of Culex tarsalis mosquitoes with RVFV strains results in efficient viral reassortment”.
Comment 1: This is an impactful manuscript describing relative rates of RVFV genomic segment reassortment in mosquito cells in vitro and in tissue types in whole animals. The authors have followed up on previous in vivo experimentation to examine reassortment of viral segments from different strains of RVFV when sheep are co-infected with multiple strains. Here, they use mosquito cells and live mosquitoes to show that rates of reassortment are dramatically higher in specific tissue types in live mosquitoes than in insect cells in culture or in mammalian cells. While there are several caveats and limitations to this study, it provides a new appreciation for propensity for genome segment reassortment in different host/vector species and in different tissue types within the mosquito.
Response 1: We thank the reviewer for their assessment of the merits of the current submitted work. We agree the data provided here achieve the goal of developing tools to investigate segmental reassortment among strains of RVFV. We are also excited for future work that will come from these experiments that will further expand our knowledge of viral reassortment.
Comment 2:It should be noted that in a recent development in the viral taxonomic structure, the old “Bunyavirales” Order has been elevated to a Class called the “Bunyaviricetes”, with two new Orders. The relevant Order for RVFV is now “Hareavirales” and the family is the “Phenuivirdae” (https://doi.org/10.1128/jvi.01069-24). The authors should make the appropriate corrections on Lines 43 and 44.
Response 2: We thank the reviewer for correcting this important taxonomic information. We have amended the text in the revised manuscript (Lines 43-45) to reflect the correct information which now reads as:
“Rift Valley fever phlebovirus (RVFV) (Order: Hareavirales; Family: Phenuiviridae; Genus: Phlebovirus) is a zoonotic pathogen capable of causing outbreaks resulting in substantial impact on human and livestock populations leading to economic loss [1].”
Comment 3:There is a mistake in describing the coding strategy of the ambisense S segment of RVFV. It says, starting on line 69: “…Small segment (S; 1.7kb) encoding a nucleoprotein (N) in the sense orientation and the non-structural protein (NSs) in the antisense orientation.” In fact (as the authors surely know), the genomic RNA on the S segment is in the negative sense (antisense) with respect to N, i.e., when transcribed you get the + sense mRNA. The NSs gene is in the positive sense in the genomic RNA, which needs to be replicated into the antigenome before transcription of the NSs gene into the + sense mRNA. This should be clarified.
Response 3: We thank the reviewer for this comment. We have added an additional citation and amended the abovementioned text in the revised manuscript to read:
Lines 67-71: “The RVFV genome consists of a Large segment (L; 6.4 kb) encoding viral RNA-dependent RNA polymerase, Medium segment (M; 3.8 kb) encoding envelope glycoproteins Gc and Gn, NSm, and the 78kDa protein, and a Small segment (S; 1.7 kb), which is ambisense and encodes a nucleoprotein (N) in the genomic-sense orientation and the non-structural protein (NSs) in the antigenomic orientation [14,15].”
The additional citation is from Gaudreault et al. 2019: Gaudreault, N.N.; Indran, S.V.; Balaraman, V.; Wilson, W.C.; Richt, J.A. Molecular Aspects of Rift Valley Fever Virus and the Emergence of Reassortants. Virus Genes 2019, 55, 1–11, doi:10.1007/s11262-018-1611-y.
Comment 4: Although the experimental design is solid and the results largely support the conclusions, one aspect that warrants more discussion and future experimentation is the lack of clear discrimination between the likely drivers of reassortment versus ‘non-reassortment’. That is, reassortment absolutely requires co-infection of individual cells. On the other hand, non-reassortment could be the result of either absence of co-infection of individual cells, or some other host cell restriction of certain combinations of segments or simply that certain reassortants exhibit significantly reduced fitness in at least one cell type in the host.
Response 4: We thank the reviewer for this observation surrounding our experimental design. We believe that we have addressed these concepts with the following text present in our discussion:
Lines 374-375: “These outcomes could be due to low levels of individual cells co-infected with different virus strains.”
Lines 387-393: “Future work can focus on altering viral infection dynamics in vitro to observe downstream patterns of reassortment. Our previous and supplemental data do not indicate a significant difference in the replication kinetics among the three strains used in our studies, supporting the use of equivalent MOIs for co-infection of the cell cultures [29; and Supplementary Fig. 2]. We selected a lower MOI to allow for multiple rounds of viral replication to increase the likelihood for co-infection by 3 days post-infection without significant cell death [20].”
Lines 452-457: “A limitation to data generated in the current study is the possibility that certain combinations of reassortants not detected with in vitro and in vivo experiments could display a slower rate of replication and/or fitness. Thus, plaque isolation of such phenotypes would not have occurred with the methods utilized here. This could be defined in future studies by assessing plaque isolation from co-infected tissue beyond the standard protocol timeline.”
Comment 5: Therefore, it is somewhat puzzling why a low MOI was chosen for the experiments in cell culture. It is strictly possible that so few cells were productively infected (with both viruses) in the cell culture experiments that reassortment did not have the opportunity to happen, while infection dynamics in a live mosquito midgut would have been more conducive numerically to coinfection and thus reassortment. On the other hand, if there were host cell type specific restriction mechanisms for which certain components of the virus strain specific segments provided a means for viral escape, this could be deduced in further in vitro and in vivo viral replication experiments.
Response 5: We thank the reviewer for this comment. We feel our rationale guiding the MOI used for the current work is best justified by the amended discussion in the revised manuscript:
Lines: 391-393:“We selected a lower MOI to allow for multiple rounds of viral replication to increase the likelihood for co-infection by 3 days post-infection without significant cell death [20].”
Comment 6: Along these same lines, although it is somewhat obvious that the authors used Vero cells for the mammalian cell infections to maximize replication/propagation, since these cells do not mount an interferon defense to infection and the RVFV countermeasures to interferon pathway activation is in S or distributed between S and M, it seems a missed opportunity that only these mammalian cells were used. The authors should discuss this, and probably put it on the list of experiments to do for next time!
Response 6: Lines 426-433: “Not investigated in the current study is the interferon response, which is suppressed by NSs during RVFV replication. Vero cells are deficient in interferon expression, which could have impacted the replication kinetics of the reassortant viruses investigated in this study [38]. It is of interest and the aim of future studies to assess the replication kinetics of the RAVs in interferon competent RVFV susceptible cell lines such as Baby hamster kidney (BHK-21), Madin-Darby ovine kidney (MDOK), or Aedes albopictus (C6/36 and/or U4.4) cells [20, 35]. In our previous work, we have used Madin-Darby ovine kidney (MDOK) cells to assess reassortment and this tool would fit well within the repertoire of resources used in the present work [20].”
Comment 7: Overall this is a good and important work.
Response 7: Thank you for taking the time to review our manuscript and for the constructive comments.

Reviewer 2 Report
Comments and Suggestions for Authors
Mosquito can transmit multiple viruses, making the co-infection of different viruses a possible scenario. Evidence has shown that some of the virus combinations can infect and thrive in the mosquito body without major fitness loss. While cases of competition or inhibition were also reported, indicating the interaction among mosquito borne viruses can be diverse and very likely viral strain and mosquito species dependent. In the current study, these authors presented co-infection study of different RVFV strains in cells as well as mosquito. Reassortment of viral segments in the co-infection cells and mosquito were determined. Replication dynamics between the reassortment viruses and the parental strains were also compared by titration while no significant replicative advantages of the reassortment strains was observed. This is an interesting study providing results of co-infection between RVFV strains, and possible reassortment of viral segments. While there are some major concerns regarding the methodology and data interpretations, which should be addressed before consideration for publication.
1) How to define the reassortment? Which regions of the three segments were pcr determined? How many pairs of primers were used for the plague typing? And if there were confirmed by sequencing or only pcr produce size? This key info regarding to the methods verification should be presented in the main text or in the supplementary materials.
2) It seems that there is a strong replicative fitness advantage of Kenya-12B-15 over the other two strains in the co-infection scenario, especially in which with SA01-1322 (figure 1). I do not think the kenya12b and SA01 is a proper couple for reassortment study since one of the virus strains is completely outcompeted leaving no room for segments reassortment. The same concern goes to the mosquito experiments, where only very few samples tested the parental SA01 meaning the same competition observed in cell study (figure 2C).
Plus, there is a gap between cell fitness and infection in mosquito, so the authors need to better provide that the SA01 strain used in this study can be efficiently transmitted by the culex mosquito strains. This will directly influence the conclusion made in Results 3.2 for I see only 1 sample of reassortment in the salivary glands in Fig 2C.
3) In mosquito co-infected with Kenya-12B-15 and MP-12 where no parental Kenya-12B-15 segment can be detected in the salivary glands samples (Figure 2B). How to explain this results? Considering that the Kenya-12B-15 is more competitive than the MP-12 in cells, it is very surprising that Kenya-12B-15 is out competed in the co-infected mosquito salivary glands. For the mosquito experiments, I strongly recommend the author to provide data of single infectious blood meal of the parental strains.
Author Response
Reviewer 2 comments:
Comments and Suggestions for Authors
Comment 1: Mosquito can transmit multiple viruses, making the co-infection of different viruses a possible scenario. Evidence has shown that some of the virus combinations can infect and thrive in the mosquito body without major fitness loss. While cases of competition or inhibition were also reported, indicating the interaction among mosquito borne viruses can be diverse and very likely viral strain and mosquito species dependent. In the current study, these authors presented co-infection study of different RVFV strains in cells as well as mosquito. Reassortment of viral segments in the co-infection cells and mosquito were determined. Replication dynamics between the reassortment viruses and the parental strains were also compared by titration while no significant replicative advantages of the reassortment strains was observed. This is an interesting study providing results of co-infection between RVFV strains, and possible reassortment of viral segments. While there are some major concerns regarding the methodology and data interpretations, which should be addressed before consideration for publication.
Response 1: We are grateful for the reviewer’s thoughtful assessment of our work. We agree there are many aspects of viral interactions in mosquitoes, particularly those co-infected and are excited to explore these aspects in future study. We hope to address concerns with the comment-specific responses provided below.
Comment 1a: How to define the reassortment?
Response 1a: We thank the reviewer for highlighting a necessary expansion of the definition of reassortment. We have amended the text of our Introduction section as stated below:
Lines 62-64: “Reassortment or viral segmental exchange during cellular co-infection is a hallmark amongst viruses within the order Bunyavirales, which possess three genome segments that are non-selectively packaged within a given host cell [11–13].”
Comment 1b: Which regions of the three segments were pcr determined? How many pairs of primers were used for the plague typing? And if there were confirmed by sequencing or only pcr produce size? This key info regarding to the methods verification should be presented in the main text or in the supplementary materials.
Response 1b: We thank the reviewer for their critical assessment of our methods. The details of the assay used to determine viral phenotype have been previously published by the co-first author under the title: RT-qPCR genotyping assays for differentiating Rift Valley fever phlebovirus strains. DOI: 10.1016/j.jviromet.2023.114693. We have cited this publication in our methods which is described below.
Lines 172-182: “A previously described One-step RT-qPCR genotyping assay was performed to determine RVFV segment composition as previously described [30]. Briefly, the genotyping assay primers were mixed with 10 µL of q-script XLT 2x mix (Quantabio, Beverly, MA, USA), 1 µL of Eva green (20x) (Biotium, Fremont, CA, USA), 2.5 µL of RNA template and nuclease free water up to a total volume of 20 µL. The amplified PCR products were subjected to one round of melt curve analysis with increasing temperatures from 70 to 95°C at the rate of 0.20°C change every 10 seconds. Melt curves were analyzed using CFX 3.0 software (Bio-Rad, Hercules, CA, USA). Based on the peak melting temperature, the samples were categorized as Kenya-128B-15 or MP-12 or SA01-1322 strains L or M or S segment. Moreover, the strain identities of genotyped plaques (approx. 10%) were confirmed by Sanger sequencing [30].”
Comment 2a: It seems that there is a strong replicative fitness advantage of Kenya-12B-15 over the other two strains in the co-infection scenario, especially in which with SA01-1322 (figure 1).
Response 2a: We thank the reviewer for this observation. We feel that our data demonstrate that in our hands there is no replicative advantage among any of three strains used here. This is directly demonstrated between Kenya-128B-15 and SA01-1322 in the growth curves included in our Supplementary Figure 2. Furthermore, viral replication curves presented in Figures 4 and 5 include parental strain data. These data depict similar growth curves between the pairings of Kenya-128B-15 and MP-12 and Kenya-128B-15 and SA01-1322. Thus, we feel our data are not the direct result of one strain outcompeting a second during the co-infection and dissemination process. We agree a key aspect of investigating reassortment is understanding the replication of each viral strain involved. We feel this would be well-investigated in future work, where viral input could be altered as an enhancement of the current work where virus was inoculated in a 1:1 ratio. We believe we have incorporated this concept in our discussion material:
Lines 387-391: “Future work can focus on altering viral infection dynamics in vitro to observe downstream patterns of reassortment. Our previous and supplemental data do not indicate a significant difference in the replication kinetics among the three strains used in our studies, supporting the use of equivalent MOIs for co-infection of the cell cultures (Supplementary Figure 2) [29].”
Comment 2b: I do not think the kenya12b and SA01 is a proper couple for reassortment study since one of the virus strains is completely outcompeted leaving no room for segments reassortment. The same concern goes to the mosquito experiments, where only very few samples tested the parental SA01 meaning the same competition observed in cell study (figure 2C).
Comment 2b: We thank the reviewer for this observation, however, we have previously used these strains for co-infection and reassortment studies in sheep and our goal was to further assess the influence of mosquitoes on this process. We understand the reviewers concern in observing only a small amount of salivary gland-based reassortment in mosquitoes co-infected with Kenya-128B-15 and SA01-1322. However, we feel the observation of reassortment in the midgut implies viral co-infection. Although, we observe a lower level of reassortment in the salivary glands this could be due to multiple other factors, such as lowered competence for dissemination among reassortants. Additionally, it is possible reassortant virus resulting from this combination of reassortment may possess depressed replication. Had we altered our viral isolation methods to include collection of plaques outside the normal RVFV window, it is possible we would have detected additional reassortants in the salivary glands and overall. We reflect these concepts in our amended discussion:
Lines 452-457:“A limitation to data generated in the current study is the possibility that certain combinations of reassortants not detected with in vitro and in vivo experiments could display a slower rate of replication and/or fitness. Thus, plaque isolation of such phenotypes would not have occurred with the methods utilized here. This could be defined in future study by assessing plaque isolation from co-infected tissue beyond the standard protocol timeline.”
Comment 2c: Plus, there is a gap between cell fitness and infection in mosquito, so the authors need to better provide that the SA01 strain used in this study can be efficiently transmitted by the culex mosquito strains. This will directly influence the conclusion made in Results 3.2 for I see only 1 sample of reassortment in the salivary glands in Fig 2C.
Response 2c: We thank the reviewer for this observation. We feel the data showing presence of the SA01-1322 strain in midgut and salivary glands demonstrates the suitability of this strain being used with Culex tarsalis. Additionally, in data not presented here, we were able to detect SA01-1322 in legs and wings collected alongside midgut and salivary tissue. We understand this does not translate to viral transmission, however, this can be remedied in future study we will undertake.
Comment 3: In mosquito co-infected with Kenya-12B-15 and MP-12 where no parental Kenya-12B-15 segment can be detected in the salivary glands samples (Figure 2B). How to explain this results? Considering that the Kenya-12B-15 is more competitive than the MP-12 in cells, it is very surprising that Kenya-12B-15 is out competed in the co-infected mosquito salivary glands. For the mosquito experiments, I strongly recommend the author to provide data of single infectious blood meal of the parental strains.
Review 3: We thank the reviewer for their insight into our experimental design. We feel given the data presented here the Kenyan strain of RVFV does not display an overt replicatory advantage over MP-12. This, however, is a concept worthy of future investigation. We are also curious to further understand the patterns of non- and reassortment observed here. It is possible several factors that are host- and/or virus segment-based may influence these outcomes. Additional time points should be incorporated to more thoroughly capture trafficking by each viral strain and reassorted products. We are currently planning future experiments and would like the opportunity to understand RVFV-based segmental factors that may dictate these reassortment patterns.

Round 2
Reviewer 2 Report
Comments and Suggestions for Authors
In the current revision, the authors have provided responses to my previous comments accordingly. I am satisfied with the responses regarding to the reassortment determination and segments test. It is bit pity that the authors decided not to provide extra mosquito experiments as I recommended which I still think would make the current story more solid and complete. But I can also understand that the decision for future study. Except the above points, I only have few that would like to discuss bit further.
As to my previous comments 2a, the authors responded that no significant replicative advantage among the three tested stains can be seen according to the viral growth kinetics, which I agree with, only in case of single infection. While in co-infection, it shows that Keny-128B-15 segments outnumbered its counterparts. This segments dominance is less clear in cells that coinfected with Keya-128B-15 and MP-12. It indicates that a similar replicative kinetics in single infection does not necessarily reflect its fitness in the co-infection scenario, or at least in the segments number accumulation level. The segments dominance of Keyna-128B-15 in the cells co-infected with SA01 is like the elephant in the room that is too big to ignore.
As to comment 2b, the authors responded that the low saliva positive sample of reassortment between Keyna-128B and SA01 could be due to either host factors restrict the dissemination and the subsequent accumulation in salivary glands of the reassortment viruses, or the reassortment viruses have a depressed replication. I tend to believe that there is a stronger selective pressure in the MG on the reassortment virus, resulting only the parental strain can escape and disseminate to other tissues. I do not see a significant difference on the replicative kinetics between the reassortment and the parental viruses (Figure 5).
Author Response
In the current revision, the authors have provided responses to my previous comments accordingly. I am satisfied with the responses regarding to the reassortment determination and segments test. It is bit pity that the authors decided not to provide extra mosquito experiments as I recommended which I still think would make the current story more solid and complete. But I can also understand that the decision for future study. Except the above points, I only have few that would like to discuss bit further.
Response: We thank the reviewer for their contributions to improving our manuscript thus far. We agree that ideally additional experiments would enhance this study; however, due to various constraints and shift of our current focus, additional experiments are beyond the scope of this current study. We hope the responses below will prove a more satisfactory version of our manuscript.
Comment #1: As to my previous comments 2a, the authors responded that no significant replicative advantage among the three tested stains can be seen according to the viral growth kinetics, which I agree with, only in case of single infection. While in co-infection, it shows that Keny-128B-15 segments outnumbered its counterparts. This segments dominance is less clear in cells that coinfected with Keya-128B-15 and MP-12. It indicates that a similar replicative kinetics in single infection does not necessarily reflect its fitness in the co-infection scenario, or at least in the segments number accumulation level. The segments dominance of Keyna-128B-15 in the cells co-infected with SA01 is like the elephant in the room that is too big to ignore.
Response #1: We thank the reviewer for this detailed analysis of our data. We hope the below responses and modifications to our discussion address these points.
We feel that a more appropriate framing of the co-infection experiments between Kenya-128B-15 should include the viral input as determined through back-titration. This reflects a higher inoculum of Kenya-128B-15. We do not feel this detracts from the presented data and hope this amendment to our discussion is appropriate.
Lines 421-442: “Viral titration of input virus revealed a higher load of Kenya-128B-15 compared to SA01-1322. This may account for representation of Kenyan parental strain and overall segments recovered in mosquito tissues. Additionally, viral dissemination determined by assessment of infected mosquito legs and wings resulted in the detection of only parental SA01-1322. This demonstrates that both parental strains, although inoculated at different titers, were able to establish infection and downstream dissemination. Previous research by Beaty et al. demonstrated that midgut escape of reassortant virus derived from La Crosse (Order: Elliovirales; Family: Peribunyaviridae: Genus: Orthobunyavirus) and showshoe hare (Order: Elliovirales; Family: Peribunyaviridae; Genus: Orthobunyavirus) virus were diminished for those genotypes containing either LAC or SSH S segments [38,39]. The RVFV strains used here have high amino acid similarity across all segments, thus, reassorted genotypes with different segmental combinations may have less impact on tissue escape and dissemination. However, it is possible, that selective pressure on reassortant strains in the midgut limited dissemination compared to parental strains. Additional work should investigate viral interactions with mosquito tissue escape barriers or alternative mechanisms for dissemination which may govern this result [40]. Experiments wherein viral MOI can be altered (as mentioned above) and those that include serial exposure to infectious bloodmeals should be undertaken. Such experiments will help to determine if superinfection exclusion plays a role in reassortment frequency and whether this occurs focally or across the entire tissue. Further, reassortment frequency and restriction derived from co-infection in alternative mosquito species, such as Aedes aegypti, should also be explored.”
We agree that it is important to understand viral fitness under the constraints of co-infection. It is possible specific RVFV proteins are key to establishing viral infection and co-infection that could produce the observed segmental outcomes. We include discussion of the concept.
Lines 494-498:“This may be more similar to observations made for La Crosse virus reassortment, where segmental exchange was more stochastic [42]. Future experiments will use the tools established here to further interrogate robustness of the observed patterns and potential contributions by proteins expressed on specific segments that may have downstream implications for RAVs detected in the field.”
Comment 2: As to comment 2b, the authors responded that the low saliva positive sample of reassortment between Keyna-128B and SA01 could be due to either host factors restrict the dissemination and the subsequent accumulation in salivary glands of the reassortment viruses, or the reassortment viruses have a depressed replication. I tend to believe that there is a stronger selective pressure in the MG on the reassortment virus, resulting only the parental strain can escape and disseminate to other tissues. I do not see a significant difference on the replicative kinetics between the reassortment and the parental viruses (Figure 5).
Response #2: Precise cellular MOI input is a key component of conducting co-infection work with reassortment being the resulting readout. This is especially challenging to control with RVFV due to the need to work with freshly-grown virus during experiments, so we can only confirm the MOI after the experiment is completed. Data shown in Supplementary Figure 1 suggest a higher input of Kenya-128B-15 for in vivo co-infection experiments with the SA0-1322 strain, impacting co-infection dynamics. This should be addressed in future work where MOI of input virus is altered between co-infecting strains to determine impact on cellular infection and reassortment frequency. These concepts are reflected in our updated discussion section.
Lines 421-442: “Viral titration of input virus revealed a higher load of Kenya-128B-15 compared to SA01-1322. This may account for representation of Kenyan parental strain and overall segments recovered in mosquito tissues. Additionally, viral dissemination determined by assessment of infected mosquito legs and wings resulted in the detection of only parental SA01-1322. This demonstrates that both parental strains, although inoculated at different titers, were able to establish infection and downstream dissemination. Previous research by Beaty et al. demonstrated that midgut escape of reassortant virus derived from La Crosse (Order: Elliovirales; Family: Peribunyaviridae: Genus: Orthobunyavirus) and showshoe hare (Order: Elliovirales; Family: Peribunyaviridae; Genus: Orthobunyavirus) virus were diminished for those genotypes containing either LAC or SSH S segments [38,39]. The RVFV strains used here have high amino acid similarity across all segments, thus, reassorted genotypes with different segmental combinations may have less impact on tissue escape and dissemination. However, It is possible, that selective pressure on reassortant strains in the midgut limited dissemination compared to parental strains. Additional work should investigate viral interactions with mosquito tissue escape barriers or alternative mechanisms for dissemination which may govern this result [40]. Experiments wherein viral MOI can be altered (as mentioned above) and those that include serial exposure to infectious bloodmeals should be undertaken. Such experiments will help to determine if superinfection exclusion plays a role in reassortment frequency and whether this occurs focally or across the entire tissue. Further, reassortment frequency and restriction derived from co-infection in alternative mosquito species, such as Aedes aegypti, should also be explored.”
The reassortant viruses used for replication curves were those most highly represented in our data. That they showed no remarkable replication differences only extends to these strains as other detected reassortants may possess altered growth profiles not investigated here. This is reflected in our discussion (Lines 475-480):
“A limitation to data generated in the current study is the possibility that certain combinations of reassortants not detected with in vitro and in vivo experiments could display a slower rate of replication and/or fitness. Thus, plaque isolation of such phenotypes would not have occurred with the methods utilized here. This could be defined in future studies by assessing plaque isolation from co-infected tissue beyond the standard protocol timeline.”